# FMM-Head: Enhancing Autoencoder-based ECG anomaly detection with prior knowledge

## Abstract

Detecting anomalies in electrocardiogram data is crucial to identifying deviations from normal heartbeat patterns and providing timely intervention to at-risk patients. Various AutoEncoder models (AE) have been proposed to tackle the anomaly detection task with machine learning (ML). However, these models do not consider the specific patterns of ECG leads and are unexplainable black boxes. In contrast, we replace the decoding part of the AE with a reconstruction head (namely, FMM-Head) based on prior knowledge of the ECG shape. Our model consistently achieves higher anomaly detection capabilities than state-of-the-art models, up to 0.31 increase in area under the ROC curve (AUROC), with as little as half the original model size and explainable extracted features. The processing time of our model is four orders of magnitude lower than solving an optimization problem to obtain the same parameters, thus making it suitable for real-time ECG parameters extraction and anomaly detection.

## 1 Introduction

Cardiovascular conditions are the main causes of death worldwide (Kaplan Berkaya et al., 2018). Tools such as electrocardiogram (ECG) measurements are utilized to monitor and identify these conditions. An ECG records the heart activity by detecting electrical signals. Electrodes positioned on different parts of the body measure the signal propagation through different planes (*i.e., leads*), thus allowing the analysis of multiple heart sections. Collecting ECG data is standard procedure for both hospitalized patients and outpatients since it allows detection of various cardiovascular conditions, such as myocardial infarction and arrhythmia. In recent years, the amount of available ECG data has increased considerably due to the availability of new data sources. Given the vast amount of available data, (*deep learning (DL)*) has been extensively employed to tackle multiple ECG-related tasks. In this paper, we propose to include ECG prior knowledge in neural networks to increase the detection of anomalies in ECG data and, at the same time, enhance explainability.

Three types of sources are driving the rapid increase in ECG data that needs to be processed. The first of these is smartwatches, such as Apple watches (Apple Inc., 2018) and Fitbit (Google, 2013) while wearable smart textiles (Nigusse et al., 2021) provide continuous and long-term ECG recording. The increasing adoption of smart, low-powered, ECG-capable devices produces a huge quantity of data, but moves the bottleneck from *monitoring* to **processing** the collected data. A second data source is the large shared databases of ECG signals. Institutions and governmental bodies are establishing digital spaces for health data to provide citizens access to their health records as well as supplying de-identified health data to companies for secondary use. Given access to these new data resources, it is expected that both foundational and clinical research will improve care processes by increasing precision in both measurement and downstream mapping onto patients. Thirdly, continuous ambulatory monitoring of high-risk patients produces a huge quantity of data, whose analysis can be difficult since it requires expert knowledge of cardiac conditions and their related effect on ECG measurements (Sampson, 2018a;b).

Anomaly detection through deep learning and (*ML*) models is a promising technique to improve care by spotting health records that deviate from the patterns of normal data *without* any knowledge of what the underlying conditions might be[1]. AutoEncoders (AEs) are a family of ML models

---

[1]In contrast, ML *classification* requires labeled data from different health conditions (*i.e.,* classes) that are used to train the model.

that are trained to be able to reconstruct the original input signal. AEs are trained only on data which show no anomaly, so that during the testing and inference phases an anomaly alert will be raised if the input sample does not belong to the normal class. Since loss functions in AEs depend on the difference between the original and reconstructed data, one could infer the presence of an anomaly by looking at the reconstruction error (Hinton & Salakhutdinov, 2006). Specifically, an anomaly can be detected when the reconstruction loss is considerably higher than in the normal case. Multiple rule-based ECG anomaly detection methods have been proposed (Bortolan et al., 2021). Unlike ML models, these techniques rely on extracting well-known parameters that are indicators for specific heart conditions. However, these methods lack generalization capabilities since they rely on strong *a priori* knowledge of what these parameters are; therefore, these assumptions hinder their usability for anomaly detection of *unknown diseases*, *i.e.,* there is no *a priori* knowledge of them.

Although the most prominent strength of AEs is the lack of assumptions regarding the classes and shapes of different inputs, the inclusion of *a priori* information about the structure of input data may be beneficial for the learning procedure. While ECG signals demonstrate different patterns depending on the underlying heart condition, their shape is composed of five waves (shown in Figure 1a), which correspond to different instants of the heart's electrical signal, as measured via the electrodes. For different heart conditions, the shape of these waves change, but the number of waves and their general structure are steady. This *weak a priori* knowledge is valid for almost all ECG classes, but this knowledge is currently not exploited by state-of-the-art anomaly detectors.

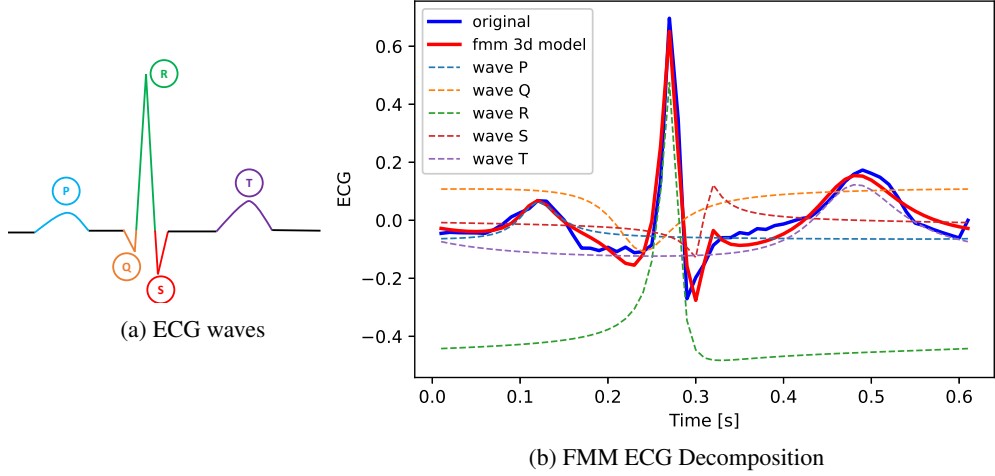

(a) ECG waves

(b) FMM ECG Decomposition

Figure 1: (a) shows the ECG shape, while (b) shows the FMM decomposition of an ECG wave

Recently, Rueda et al. (2019) proposed *Frequency Modulated Möbius (FMM)* waves to provide explainable parameters for ECG data (Rueda et al., 2022). They proposed an optimization algorithm to iteratively compute the amplitude, position, direction, and frequency parameters for the five waves composing the ECG signal through a cycle of polarization and depolarization. However, this optimization takes tens of seconds to be solved for a single heartbeat, thus making it unsuitable for real-time monitoring of critical patients and processing of voluminous quantities of ECG data. Yang et al. (2022) have shown that a neural network (NN) can be used to approximate the FMM coefficients and correctly classify heartbeats, but did not apply it for anomaly detection.

Our contributions are threefold. Firstly, we develop FMM-Head, a first approach for incorporating *weak a priori* knowledge of the ECG leads' structure into an AE model. In particular, FMM-Head replaces the decoding sub-model of AEs, provides an explainable representation of the FMM parameters, and reconstructs the signal accordingly (see Figure 1b). We design a generic pooling layer to handle the difficult task of adapting FMM-Head to different dimensions of hidden representations coming out of the encoding part of AEs. Secondly, we demonstrate FMM-Head's ease of use by incorporating it into five baseline AEs models, thus showing that our layer is flexible enough to handle the output of different kinds of encoders. FMM-Head significantly enhances the performance of these AEs. Moreover, as shown in Figure 2, even low-performing models such as EncDecAd can be enhanced to be on par with other models. Thirdly, we evaluate and compare our enhanced models to

the baselines. Replacing the decoder with the FMM-Head almost halves the total number of trainable parameters of the AEs and leads to up to $-77\%$ reduction in inference time and $-47\%$ memory required to store the model. Using a fully connected AE with 6 layers, the execution time is 21 thousand× lower than the optimization solution to the FMM problem using the Rueda et al. (2019) code, which was not designed to perform anomaly detection. The 4 orders of magnitude lower time to process batches of heartbeats enables real-time anomaly detection and is suitable for analyzing huge amounts of ECG data. Our model also provides coefficients that are highly correlated with the original FMM coefficients, thus making its output more transparent than blackbox AEs, whose extracted features are not explainable. Although training for anomaly detection reduces the output similarity to the real FMM coefficients, it allows improving the *detection of anomalies* compared to five baseline models.

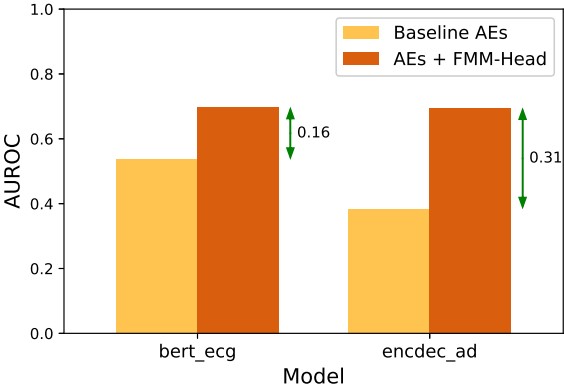

Figure 2: Anomaly detection improvement for a transformer model (0.16) and a Long-short term memory (LSTM)-based model (0.31) when augmented by FMM-Head.

## 2 BACKGROUND

### 2.1 ELECTROCARDIOGRAMS

For over a hundred years, ECGs have been used to detect heart conditions, such as myocardial infarction and arrhythmia (AlGhatrif & Lindsay, 2012). The main idea behind ECG monitoring is to repeatedly measure the electrical polarization and depolarization waves that propagate through the heart muscles and cause rhythmic contraction and relaxation. A standard 12-lead ECG machine utilizes 10 electrodes, which can be combined in different pairwise combinations to measure the voltage through different planes that intersect the heart with different orientations, thus giving insights into various parts of the heart (*e.g.,* inferior, superior anterior, posterior leads). In contrast, smartwatches produce one lead of ECG data, *i.e.,* the plane intersecting the heart through the arm, similarly to having 2 electrodes.

Figure 1a shows the shape of an ECG, composed of 5 waves corresponding to rhythmic polarization and depolarization phases. The P wave corresponds to the depolarization of the atria. The QRS complex depends on depolarization of ventricula before their contraction, whereas the T wave is determined by ventricular repolarization. Detecting a cardiac disturbance in conductance may involve different features of the ECG, such as the interval and slope between the peaks, their amplitude or underlying area. Depending on the task, a cardiologist employs this information to identify conditions and diseases. Instances of such conditions with relative ECG change are nodal tachycardia (hidden P-wave), sinus tachycardia (visible P wave with higher rate than normal), hypertrophic cardiomyopathy (deep and narrow Q waves in specific leads), myocardial infarction (ST segment elevation or depression), and atrial fibrillation (wide QRS complex, absence of P wave).

Given the bulk of possible heart conditions and their corresponding relevant ECG features, anomaly detection and classification of ECG data requires extensive knowledge (Kaplan Berkaya et al., 2018). The problem is further exacerbated when doing real-time monitoring of patients (Sampson, 2018a;b); hence, it requires automatic processing to scale with the quantity of available data.

## 2.2 FMM WAVEFORMS

To provide a comprehensive way to parametrize the rhythmic behavior of the hearth, Rueda et al. (2022) proposed modeling each individual heartbeat as the sum of 5 FMM waves (Rueda et al., 2019). Each wave is modeled as per the following equation:

$$
\begin{align}
W(t_i) &= \mu(t_i) + e(t_i) = M + A\cos(\varphi(t_i)) + e(t_i) \tag{1}\\
\varphi(t) &= \beta + 2\arctan(\omega\tan(t - \alpha)) \tag{2}\\
\mathbf{e} &= (e(t_1), \ldots, e(t_n))^T \sim \mathcal{N}(\boldsymbol{e}; 0, \sigma^2\boldsymbol{I}) \tag{3}
\end{align}
$$

where, $A \in \mathbb{R}^+$ is the amplitude of the wave, $\alpha \in [0, 2\pi)$ represents the position of the peak, $\beta \in [0, 2\pi)$ is the peak direction, $\omega \in [0, 1]$ parametrizes the lobe width of the peak, $M \in \mathbb{R}$ is the constant offset of the wave, $t_i$ is the timestep index. The tuple $\theta = (A, \alpha, \beta, \omega, M)$ represents the encoded parameters necessary to represent each wave. Their proposed 3D model includes the FMM waves formulation but makes assumptions of the parameters that are shared between leads, *i.e.,* the $\alpha$ and $\omega$ coefficients are shared among leads while $A$, $\beta$, and $M$ are not. Therefore, the final lead vector $\boldsymbol{X}$ is, for each lead L:

$$
\boldsymbol{X}(t_i)^L = M^L + \sum_{j\in\{P,Q,R,S,T\}} W(t_i, A_j^L, \alpha_j, \beta_j^L, \omega_j) + e^L(t_i); \tag{4}
$$

To provide an estimate of the optimal parameters, the following objective function is used:

$$
\theta^* = \min_\theta \sum_L \sum_i (\boldsymbol{X}^L(t_i) - \hat{\boldsymbol{X}}^L(t_i, \theta))^2; \tag{5}
$$

The estimated best tuple $\theta^*$ is obtained by repetitively iterating a fitting and wave assignation phase. During fitting, optimization is performed over a single FMM wave, and the single-lead parameters are extracted by solving a linear regression problem. More than 5 waves might be obtained. During wave assignation, a choice of the best ones is made: each peak (P, Q, R, S, T) is selected based on the $\alpha$ coefficient and thresholds on the main model's parameters. The proposed algorithm is inherently sequential, not parallelizable and it often requires minutes-order of magnitude of execution time.

## 2.3 AUTOENCODERS

AEs (Figure 3a) are a family of self-supervised NN models (Hinton & Salakhutdinov, 2006). AEs usually have a dumbbell structure (*i.e.*, wide-narrow-wide as in Figure 3a), sequentially combining $(i)$ an encoder, which transforms the inputs' features into a lower-dimensional latent representation, and $(ii)$ a decoder, that reconstructs the input from the latent representation. The loss function usually computes the error between original and output data. Hence, the aim of an AE is to exactly reconstruct the inputs. However, due to the dumbbell shape, irrelevant information is lost during inference, thus enforcing a compact, semantically meaningful latent space.

AEs are state-of-the-art models for anomaly detection (Chalapathy & Chawla, 2019). AEs are trained on normal data so that abnormal samples during test phases can straightforwardly be recognized. Samples unseen during training, so-called holdout data, will be wrongly encoded and decoded, thus causing a large loss. Different kinds of NN models have been proposed to tackle ECG anomaly detection, including LSTM-based models (Chauhan & Vig, 2015; Roy et al., 2023; Liu et al., 2022; Malhotra et al., 2016), transformers (Alamr & Artoli, 2023) and variational AEs (Jang et al., 2021).

## 2.4 CIRCULAR VARIABLES

A representation of circular data (Lee, 2010) is necessary whenever the direction of a measure is a crucial feature to understand the correspondent phenomenon. ECGs are intrinsically circular since the electrical signal passing through the heart is quasi-periodic and can be modeled as an oscillator. In the FMM formulation, $\alpha$ and $\beta$ are circular variables since they respectively represent a position and a direction within the $[0, 2\pi]$ interval.

The circular mean of a circular random variable $\theta$ can be computed as: $\bar{\theta} = \arctan 2 \left( \sum_{i=1}^{n} \sin(\theta_i), \sum_{i=1}^{n} \cos(\theta_i) \right)$. The correlation between circular variables should be computed differently than linear correlation. For instance, the linear Pearson coefficient $\rho$ between two random variables $x$ and $y$, for whom we extracted n samples, can be computed as: $\rho = \frac{\sum_{i=1}^{n}(x_i - \bar{x})(y_i - \bar{y})}{\sqrt{\sum_{i=1}^{n}(x_i - \bar{x})^2 \sum_{i=1}^{n}(y_i - \bar{y})^2}}$. However, using the Pearson coefficient between two circular variables will return an incorrect estimate of the correlation. For example, a realization of $x, y = \epsilon, 2\pi - \epsilon$ should positively contribute to the correlation coefficient. To solve this problem, circular correlation (Jammalamadaka & Sarma, 1988) can be computed by using the circular mean instead of the linear mean: $\rho = \frac{\sum_{i=1}^{n} \sin(\theta_{1i} - \bar{\theta}_1) \cdot \sin(\theta_{2i} - \bar{\theta}_2)}{\sqrt{\sum_{i=1}^{n} \sin^2(\theta_{1i} - \bar{\theta}_1) \cdot \sum_{i=1}^{n} \sin^2(\theta_{2i} - \bar{\theta}_2)}}$.

## 3 METHODOLOGY

Our FMM-Head layer reconstructs the original ECG input and provides the corresponding explainable FMM coefficients. To do so and maintain FMM parameter explainability, we split the training procedure into a warm-up regression phase and an anomaly detection training phase. Combined with constraints imposed inside the NN, this two-phase approach provides meaningful FMM coefficients.

### 3.1 PREPROCESSING

Pre-processing ECG data is essential to extract heartbeats and provide the correctly structured input data. We build on top of the pre-processing done in Rueda et al. (2022), whose pipeline consists of $(i)$ low pass filtering to remove baseline wandering, a common artifact in ECGs due to breathing or movement of the patient, $(ii)$ application of the Pan-Tompkins algorithm (Pan & Tompkins, 1985) to detect R-peaks, $(iii)$ extraction of ECG heartbeats by selecting the interval around the R-peak with $40\%$ of the distance from the previous peak and $60\%$ from the next one. Additionally, we zero-pad each sequence to a constant length in order to be able to feed the input samples to the evaluated models. The original coefficients of the FMM-model are also preprocessed. In particular, circular variables such as $\alpha$ and $\beta$ are split into their cosine and sine, so that they can be easily learnt by the NN. Also, the parameters are sorted according to the $\alpha$ parameter, so that the P wave corresponds to the first coefficients, etc.

### 3.2 FMM-HEAD

We design a novel layer, which we name FMM-Head since it resides on top of the NN and draws inspiration from the FMM formulation detailed in Section 2.2. As depicted in Figure 3b, the main idea behind FMM-Head is that any hidden representation encoded into the latent space can be mapped to meaningful FMM coefficients. This mapping is provided by a non-linear function that is implemented through one pooling layer and two fully connected networks, followed by suitable activation functions that produce parameters in the correct range for the FMM formulation. The parameters obtained after the activation function are used to reconstruct an ECG time-series $\hat{X}$ for anomaly detection. We leverage this weak *a priori* knowledge drawn from the FMM formulation to better reconstruct the input ECG signal.

The main drawback of standard anomaly detection through FMM waves is that the reconstruction might not reflect the original meaningful pattern of the FMM optimization. Although it is straightforward to obtain FMM waves that approximate an ECG signal, obtaining peaks with meaningful shapes is challenging. Hence, we propose an initial warm-up regression phase using the original FMM coefficients. This procedure constrains the output of the NN to be in the range of the original FMM coefficients, thus steering the anomaly detection phase to correct and meaningful patterns.

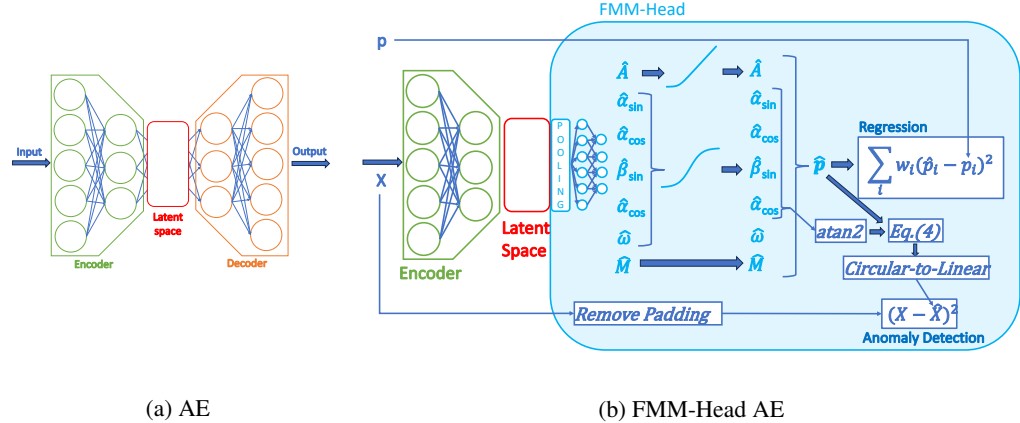

(a) AE                    (b) FMM-Head AE

Figure 3: *(a)* Structure of standard AE and *(b)* AE with the FMM-Head decoder. The FMM-Head is usually smaller than the baseline decoders.

### 3.2.1 POOLING LAYER

Depending on the employed AE, the encoder produces a latent space with different shapes. To handle different dimensions of hidden representations, a layer that maps them to a common output is needed. The pooling layers take as input the latent representation and generate a 2D output, which can be used as input to the following fully connected layers. The pooling layer is, in general, different for each encoder. Transformer and LSTM-based networks have output shapes consisting of a batch, time step, and feature dimension, which can be reduced to two by applying a linear transformation to each time step feature. For fully connected AEs and LSTM layers with state output (such as Malhotra et al. (2016)) there is no need for pooling since the output is already 2D. For convolutional networks, we flatten the 3D output space into a 2D representation.

### 3.2.2 FULLY CONNECTED LAYERS

Two fully connected layers are used to map the output of the pooling layer to the size of the FMM parameters. The first layer utilizes a non-linear function with $\tanh$ activation, while the second layer employs a linear activation. While the second layer has a fixed size (depending on the number of parameters $N$), the number of units of the first layer can be flexibly chosen. We tested multiple options, including 64, 128, and 256 units, and found only slight differences in performance. When the size is too small, there is a drastic decrease in performance. Hence, we chose one hidden layer composed of 256 units and a second layer with $N$ units for the fully connected network.

### 3.2.3 ACTIVATION FUNCTIONS FOR FMM COEFFICIENTS

The output of the fully connected network is unsuitable for direct generation of FMM waves since different parameters have different range requirements. We handle this by selecting an appropriate activation function for each parameter. In particular, a non-negative amplitude is obtained by applying *softplus*, while restricting the $\omega$ parameter to the range $(0, \omega_{max})$ with a sigmoid-like function. The sine and cosine components of the circular variables are also mapped to the $(0, 1)$ interval:

$$\hat{A}' = \ln(1 + e^{\hat{A}}), \omega' = \frac{\omega_{\max}}{1 + e^{-\omega}}, \text{ and } \alpha'_{\text{sin/cos}} = \frac{2}{1 + e^{-\alpha_{sin/cos}}} - 1.$$

The predicted $\alpha$ and $\beta$ are obtained by computing the angle that corresponds to the predicted sines and cosines. The use of $\omega_{max}$ instead of the original unitary limit reduces the possibility of non-meaningful waves with good reconstruction. With high $\omega$ values, peaks can be too wide and hence negatively influence the pattern of other peaks.

### 3.2.4 REGRESSION AND ANOMALY DETECTION

The predicted parameters can be directly used to compute the mean squared error loss. We directly inject the original FMM parameters, obtained by running FMM optimization, to compute the error

and backpropagate it to train the AE. We employ the cosine and sine values for circular parameters to compute the loss and also apply a weight to each parameter to produce better alignment with the original time series. Specifically, we apply a $10\times$ higher weight for the parameters of the R-peak to obtain well-separated waves in the QRS complex. We perform a warm-up regression phase on the original coefficients by training the AE to produce results closer to the original optimized ones. Hence, although such a network cannot be employed for anomaly detection, the output of the NN provides a prediction of the FMM coefficients.

After the warm-up regression, the NN is trained as a standard AE. We employ Equation (4) to generate the 5 ECG waves, sum them, and obtain a reconstructed signal in the $[0, 2\pi)$ domain. We then map the signal to the original length through a linear transformation and compute the mean squared error between the input ECG, stripped of the zero-padding, and the predicted sum of waves.

## 4 EXPERIMENTAL RESULTS

We evaluate FMM-Head on three datasets: Shaoxing Zheng et al. (2020), PTB-XL (Wagner et al., 2020), and ECG5000 (Chen & Keogh, 2000). We use five models as baselines: ECG-NET (Roy et al., 2023), EncDecAD Malhotra et al. (2016), a transformer model referred as BertECG (Alamr & Artoli, 2023), CVAE (Jang et al., 2021), and a fully connected AE. We employ lead 2 to train the AEs on normal data and then test the anomaly detection performance on a holdout set that includes both abnormal and normal classes.

**With warm-up regression, FMM-Head consistently enhances the anomaly detection capabilities of the baseline models by up to 0.31 of area under the ROC curve (AUROC)**. Table 1 shows the AUROC for the evaluated models and datasets and the gains compared to baselines for the different models. For the Shaoxing and PTB-XL datasets, our pipeline consists of warm-up regression followed by anomaly detection training. In these cases, the performance increase is consistent for all combinations of datasets and models. The AUROC enhancement is more evident for low-performant baselines, such as EncDecAd and BertECG, where the evaluated AUROC can increase by up to 0.31. The main reason some baselines perform poorly is the complexity of the performed task. Compared to simple datasets (such as ECG5000), PTB-XL and Shaoxing are among the largest publicly available ECG data pools, consisting of multiple patients and labeled conditions. We have experimentally determined that one of the major sources of complexity is the variable length of the ECG time series. While most models do not consider this factor, our FMM-Head inherently maps the time series to the $[0, 2\pi)$ interval, thus equalizing the lengths of the samples in the last layer. Therefore, even low-performant baselines can achieve considerable AUROC for variable-length training sets.

The benefits of FMM-Head are noticeable for high-performing baselines, such as ECG-NET and CVAE. Replacing the decoder with a head based on prior knowledge better exploits the features extracted from the encoder. One exception is FMM-CAE, whose baseline counterpart CVAE performs slightly better for the PTB-XL dataset. We argue that convolutional AEs are mostly focused on recognizing patterns between adjacent time steps, therefore being less suitable for the extraction of features for FMM coefficients. Our belief is confirmed by the AUROC value for FMM-CAE and CVAE for the Shaoxing dataset, with a 0.02 decrease in AUROC, which is the largest decrease in performance of all of the model-dataset combinations. However, for the ECG5000 dataset the AUROC is already very high and the addition of the FMM-Head makes little (at most 1.7%) difference.

**Baselines with integrated FMM-Head can extract coefficients that are highly correlated with those extracted by solving an optimization problem in less than $\frac{1}{20000}$ of the time**. Although our model is built for anomaly detection and *not* to exactly reproduce the FMM coefficients, we produce coefficients correlated with those obtained with the original FMM optimization.

Figure 4a shows the linear or circular correlation between the coefficients extracted by Rueda et al. (2022) and those obtained after warm-up. The extracted parameters are usually highly correlated (*i.e.,* value greater than $0.4$). Noteworthy is that the P and T waves manifest in general high correlation to the optimized ones. This is due to the two peaks being spaced apart by the QRS complex, thus making them more straightforward to extract. In contrast, the waves belonging to the QRS complex are close to each other, thus enabling the same reconstructed time series through possibly non-ideal wave combinations. For instance, the same ECG pattern could be obtained through the sum of three wide extracted waves instead of three narrow peaks. Therefore, the inference of the QRS complex

Table 1: Highest AUROC values for best learning rate and gains compared to baselines for different models. Our FMM-HEAD enhanced models produce consistently better results compared to the baselines when the warm-up regression is employed (*i.e.,* for Shaoxing and PTB-XL)

| | Shaoxing | | PTB-XL | | ECG5000 | |
| Model | AUROC | Gain | AUROC | Gain | AUROC | Gain |
|---|---|---|---|---|---|---|
| ecgnet | 0.575 | | 0.661 | | 0.993 | |
| fmm_ecgnet | 0.659 | **0.084** | 0.731 | **0.070** | 0.988 | *−0.005* |
| encdec_ad | 0.461 | | 0.384 | | 0.982 | |
| fmm_encdec_ad | 0.617 | **0.156** | 0.695 | **0.311** | 0.988 | **0.006** |
| bert_ecg | 0.545 | | 0.536 | | 0.971 | |
| fmm_bert_ecg | 0.650 | **0.105** | 0.697 | **0.161** | 0.989 | **0.017** |
| dense_ae | 0.653 | | 0.691 | | 0.992 | |
| fmm_dense_ae | 0.699 | **0.046** | 0.698 | **0.007** | 0.990 | *−0.002* |
| cvae | 0.749 | | 0.693 | | 0.992 | |
| fmm_cae | 0.729 | *−0.020* | 0.727 | **0.034** | 0.990 | *−0.002* |

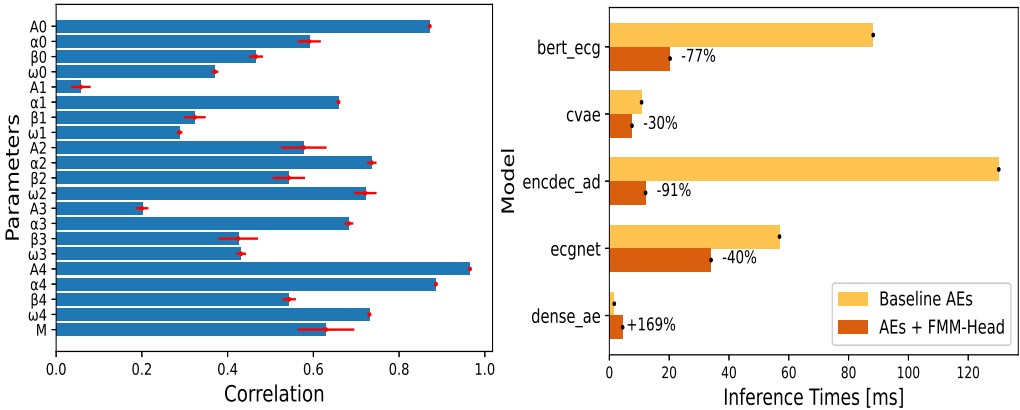

(a) Correlation between FMM coefficients for Shaoxing

(b) Inference Times for Shaoxing

Figure 4: Despite not being built for FMM coefficient extraction, FMM-Head based models produce FMM parameters that are highly correlated with those extracted by the original method (see FMM-DenseAE in *(a)*). *(b)* Compared to the baselines, our NN-based models provide lower inference, except for models with low-complexity operations such as DenseAE.

with a NN is less correlated to the original when compared to the P and T wave. The results for PTB-XL are similar to those for Shaoxing, except for the inference time gains for the LSTM-based models, which are lower due to the smaller input size.

Figure 4b shows the decrease in inference time compared to baseline models for the Shaoxing dataset. Compared to the original FMM optimization problem, which takes in average $38\,\text{s}$, AEs with FMM-Head reduces the computation time by more than four orders of magnitude. Compared to the baseline AEs, the benefits of FMM-Head varies from $-30\%$ to $-91\%$, with the exception of DenseAE. The main reason behind this is that although FMM-Head is small, it is still more complex than most state-of-the-art layers. For DenseAE, replacing the fully connected 3-layer decoder with FMM-Head does *not* reduce the inference time but actually increases it by $+169\%$. However, dense, convolutional, and LSTM layers have been extensively researched and optimized in the last decades; hence, we argue that FMM-Head execution times could be further improved by optimizing the code.

**By replacing standard decoders with FMM-Head, the model size is nearly halved for the non-LSTM evaluated models, greatly reducing the storage requirements on mobile and wearable ECG monitoring devices**. Figure 5a shows the reduction in model size obtained by replacing each baseline's decoders with FMM-Head. In most cases, the file size of the obtained models is nearly halved. The large decrease in model size on BertECG is due to a reduction in the encoder, which we experimentally showed did not produce a relevant impact on the performance of the correspondent

model integrated with FMM-Head. Noteworthy is that LSTM-based models such as ECG-NET and EncDecAD gain the least benefits from FMM-Head in terms of model size. This is due to the inherent design of LSTM layers, which favor minimizing model size over training and inference time; hence, the gains of FMM-Head are less prominent. In particular, the size of ECG-NET is 51% higher when FMM-Head is employed since the long inputs of Shaoxing increase the size of the pooling layer and consequently increase the size of the first fully connected layer.

Figure 5b shows the epoch duration during one training session of each model on the Shaoxing dataset. As expected, the reduced model size enables faster training epochs than the baselines. Similarly to Figure 4b, the only exception is DenseAE, which trades off the model size with the additional complexity of the FMM-Head, thus showing an increase in the epoch duration.

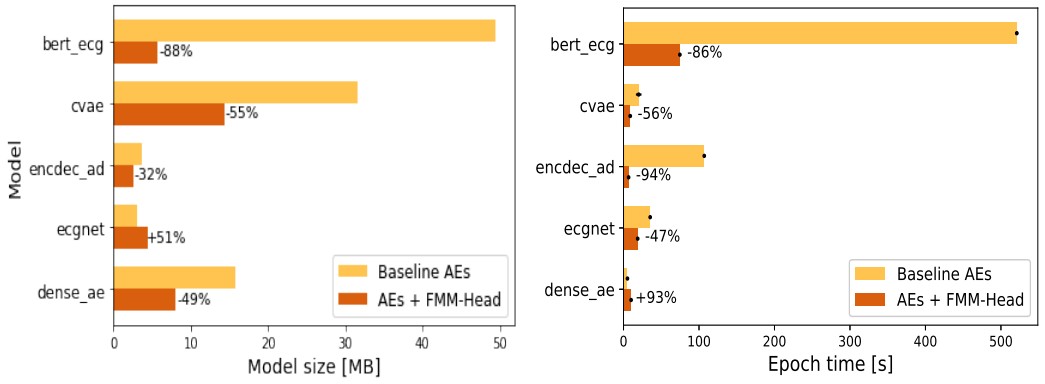

(a) Model sizes for Shaoxing dataset  (b) Epoch duration for Shaoxing dataset

Figure 5: *(a)* FMM-Head has negligible size compared to most decoders, and thus one can considerably reduce the amount of file storage and memory needed to compute training and inference for neural network models. *(b)* shows the epoch duration for different models during training.

**Warm-up regression is essential to maximize the benefits of the FMM-Head**. Column 3 in Table 1 shows the performance of different models on the ECG5000 dataset. Whereas for EncDecAd and BertECG the AUROC is slightly better, for the other baselines we notice a decrease instead. We claim that this is due to the fact that we did not perform any warm-up regression on ECG5000, thus not exploiting the prior knowledge of the ECG shape. Without warm-up, the waves are not constrained in a meaningful position and the increase in AUROC is consequently less prominent.

**Compared to Yang et al. (2022), FMM-Head can be applied to multiple baseline NNs for anomaly detection and not just classification**. Yang et al. (2022) provides a way to estimate 12-leads FMM coefficients by means of a custom NN and use them for ECG classification with Support Vector Machine (SVM) or logistic regression. Compared to it, FMM-Head can instead be applied to multiple AEs for anomaly detection instead of classification with a single model. Moreover, in Yang et al. (2022) the waves are centered into the correct position by adding a loss regularization term, which is based on 100 random samples, whose coefficients are obtained through the non-ECG specific code from Rueda et al. (2019). We instead obtain it with warm-up on *all* the coefficients, obtained from the ECG-specific R code from Rueda et al. (2022). Moreover, we explicitly consider $\alpha$ and $\beta$ as circular variables, thus allowing correct estimates of their values and correlations. However, our current implementation only handles a single lead.

## 5 CONCLUSION AND FUTURE WORKS

We introduced a novel way of inserting *a priori* knowledge into AEs for anomaly detection applied to ECG data. Our FMM-Head increased the anomaly detection capabilities of five baselines and reduced model size and inference time. Our method makes it possible to perform real-time, explainable anomaly detection for continuously monitored patients. As future work, we will investigate how FMM-Head can handle 12 leads ECG data. We will analyze how only employing the regression loss can generate coefficients that are more correlated to the original ones. The inference time of FMM-Head can be improved by better exploiting parallel operations in TensorFlow operations.

## 6 Reproducibility Statement

The employed preprocessing of the ECG signals can be found in Section 3.1. The FMM-Head pooling layer, which is, in general, different for each baseline model, is described in Section 3.2.1. The description of the employed datasets and models can be found in Appendix C. The experimental setup, hardware specification, and training time estimates are detailed in Appendix C.4. Finally, the source code will be shared on the discussion forums as a link to an anonymous repository during the review phase.

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

# A   APPENDIX

# B   DEEP LEARNING MODELS FOR TIME-SERIES

This section describes different kinds of state-of-the-art deep learning mechanisms and their applications to ECG signals. In particular, we summarize in appendix B.1 how an artificial works, and provide details of multiple kinds of NNs in the following subsections.

## B.1   ARTIFICIAL NEURAL NETWORKS (ANNS)

As noted earlier, the amount of ECG data is constantly increasing; therefore, tools for automatic ECG anomaly detection and classification have become paramount. One goal of such tools is to detect potential anomalies in the heart's electrical signals and provide explainable motivations in the shortest time possible. For these reasons, ML and DL models have been proposed for various tasks on ECG data (Kaplan Berkaya et al., 2018). Unlike rule-based approaches, ML learns to discriminate between different kinds of samples by learning from pools of available data. DL is a branch of ML which focuses on the use of a (Artificial Neural Network (ANN) or simply NN) to tackle ML tasks. NNs are composed of multiple layers of units/neurons, each of them emulating the behavior of a biological neuron. Each unit weights the inputs from the previous layer, sums them up, adds a bias term, and then passes them through a non-linear function. The NN can be trained to perform a certain task by computing the current error of the NN's output as compared to the ground truth. The backpropagation algorithm (Rumelhart et al., 1986) propagates the error backward to each layer and modifies the weighting matrices and biases to minimize the error. Many advances and kinds of NNs have been proposed in recent decades. We focus on DL models for ECG data and build on AEs, which is a NN suitable for anomaly detection.

## B.2   LSTM MODELS

LSTM models (Hochreiter & Schmidhuber, 1997) are a type of recurrent NN inherently built for time series-related tasks. LSTM layers consist of a single LSTM cell, which retains a long-term and a short-term state while the input changes according to the time step. The state is modified at each time step according to learnable gates, which select which information needs to be forgotten or inserted as the state propagated.

The main drawback of LSTM-based models is that they require backpropagation through time (BTT). BTT propagates the gradients of the loss functions back through all the time steps and is therefore not parallelizable. Hence, LSTM layers require extensive training time.

Multiple LSTM-based AEs have been proposed to tackle ECG anomaly detection. For example, Roy et al. (2023); Liu et al. (2022) stack multiple LSTM layers to reconstruct an input ECG by retaining information on the long-term state. Malhotra et al. (2016) propose a 2-layer LSTM model where the state of the last LSTM cell is used as latent space instead of the output.

## B.3   TRANSFORMERS

Transformer models are based on parallel multi-head attention and self-attention mechanisms. They were initially introduced (Vaswani et al., 2017) to handle natural language processing tasks, such as translation and sentiment analysis. Attention (Bahdanau et al., 2014) captures long-term dependencies between values of the input time-series without retaining the information in the state as in LSTM layers. Instead, each input token can attend to other tokens by simply using matrix multiplications and softmax:

$$\text{Attention}(Q, K, V) = \text{softmax}\left(\frac{QK^T}{\sqrt{d_k}}\right) V \qquad (6)$$

where Q, K, and V are the query, key, and value matrix (respectively). A context vector can be obtained through multiple self-attention layers. The context can then be used to attend to the most important parts of the input, thus making the transformer model suitable for language translation

and generation, where long-term relations between words often happen. Moreover, using only self-attention can provide remarkable performance for language classification tasks, as shown in Devlin et al. (2018).

Since attention does not take into account word/time-step order, positional encoding is often added in the initial layers of the models. The standard implementation of a positional encoding layer transforms the linear index of each word/time-step into unique sine-cosine values, thus differentiating the same words in different positions.

Transformer models have been applied for ECG anomaly detection and classification. Alamr & Artoli (2023) proposed a multi-layer transformer to detect anomalies on single heartbeat data. Gaudilliere et al. (2021) employed a transformer-like structure to classify long ECG sequences, with each heartbeat encoded as a token. Natarajan et al. (2020) proposed including handcrafted features in the transformer's output to better classify cardiac anomalies.

### B.4 Variational Autoencoders

Variational autoEncoderss (VAEs) are a variant of AEs with enhanced generative capabilities based on variational inference. In VAEs (Kingma & Welling, 2022), the latent space is modeled as encoded mean and logarithmic standard deviation of a multi-variate normal probability distribution. Unlike AEs, a sample is drawn from the latent distribution and used as input to the decoder. The loss function is a linear combination of the reconstruction error, as in AEs, and the Kullback–Leibler (KL) divergence between the latent distribution and a normal distribution with zero mean and unitary standard deviation. The presence of the KL loss constrains the model to a specific shape, thus avoiding overfitting. It also allows for formulating the problem as a statistical optimization problem. Moreover, since the latent space is probabilistic, novel reconstructed outputs can be generated simply by performing sampling multiple times. Therefore, the generative capabilities of the VAE are enhanced while still retaining its anomaly detection function.

## C    Evaluation

We evaluate the performance, inference time, and model size of our FMM-Head on three datasets and five baseline AEs from the relevant literature. After preprocessing and possibly heartbeat segmentation, we compute the FMM coefficients for the five waves. We employ lead 1 to train the AEs on normal data and then test the anomaly detection performance on a holdout set that includes both abnormal and normal classes. To obtain the best-performing model for each baseline and an FMM-Head-enhanced AE, we run multiple experiments with different learning rates in a range between $10^{-3}$ and $10^{-5}$. To prove the stability of our experiments, we ran five differently-seeded experiments per learning rate and reported the average performance and standard deviation. All seeds were saved for the purpose of replicability. If applicable, we ran the training session for 500 warm-up epochs (for VAE and baselines plus FMM-Head)followed by 500 epochs. We perform early-stopping based on the loss computed on a validation dataset. We ran our experiments in Tensorflow(Abadi et al., 2016) 2.12.0 and Python 3.8.11. The code to reproduce our results is available at Anonymous (2023). The following section describes the tested datasets (Appendix C.1), models and hyperparameters (Appendix C.2) for the FMM-Head, and evaluated metrics (Appendix C.3).

### C.1    Datasets

In the following sections we provide a description of the employed datasets. In Table 2 we report the number of total heartbeats, normal heartbeats and abnormal heartbeats after the employed preprocessing steps for the considered datasets.

Table 2: Dataset specifics

| Dataset | Num samples | Normal samples | Abnormal samples |
|---------|-------------|----------------|------------------|
| Shaoxing | 69663 | 45122 | 24541 |
| PTB-XL | 106709 | 60058 | 46651 |
| ECG5000 | 5000 | 2919 | 2081 |

### C.1.1 SHAOXING

The Shaoxing dataset (Zheng et al., 2020) contains 12-lead ECG data samples collected from $10\,646$ patients at Chapman University and Shaoxing People's Hospital. The dataset contains both abnormal rhythms and heartbeats, with 11 labeled rhythms and 67 heart conditions, including normal rhythm and absence of evident cardiovascular disease. It also includes additional data, such as age group and sex, and relevant ECG features, such as QT interval and QRS duration. The sampling frequency is $500\,\mathrm{samples/s}$, making the samples suitable for fine-grained anomaly detection. The duration of each time series is $10\,\mathrm{s}$, thus producing $5\,000$ timesteps-long samples. After preprocessing and segmentation, we zero-pad the sequences to approximately two times the heartbeat duration (*i.e.,* $1\,000$ timesteps) and remove the heartbeats that exceed this threshold.

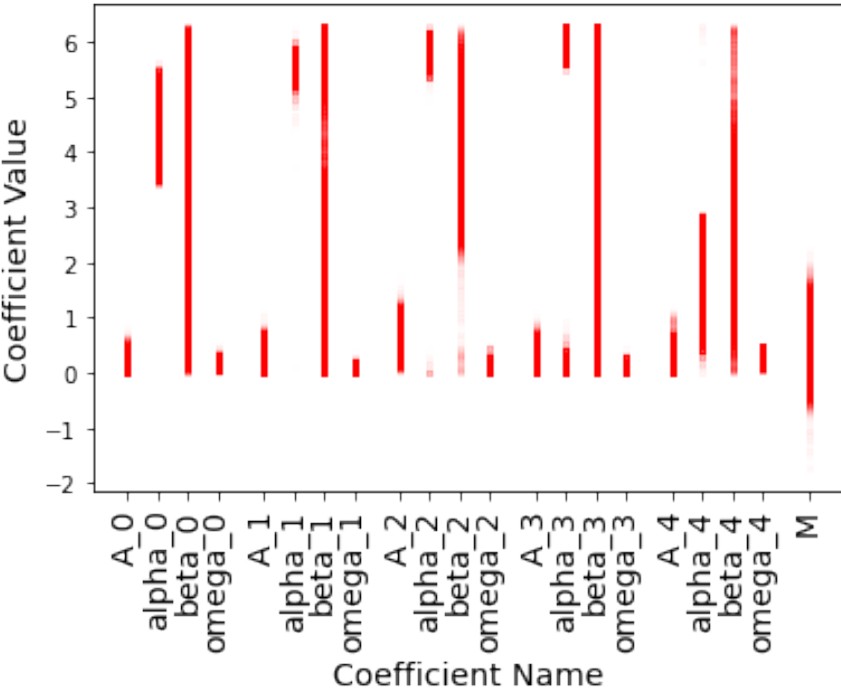

Figure 6: Coefficient distribution for the training set of Shaoxing

### C.1.2 PTB-XL

The PTB-XL dataset (Wagner et al., 2020) is the largest publicly available source of ECG data. It consists of $21\,837$ 12-leads time series from $18\,885$ patients, whose data was collected by the Physikalisch Technische Bundesanstalt (PTB) between 1989 and 1996. The samples are labeled according to 5 superclasses (Normal, Myocardial infarction, ST-T change, conduction disturbance, and hypertrophy) and a total of 28 sub-classes. Two sampling frequency versions of the datasets are available. We focus on the $100\,\mathrm{samples/s}$ one in order to provide a different angle compared to the Shaoxing dataset. The samples are already stratified in 10 slices. We use the first 9 slices in the training set and the last one in the test set. A validation dataset is drawn from the training dataset with a $0.1$ split. The duration of each sample is also $10\,\mathrm{s}$, which leads to $1\,000$ timesteps-long time series. As for the Shaoxing dataset, we zero-pad the pre-processed sequences to approximately three times the heartbeat duration (*i.e.,* 300 timesteps).

### C.1.3 ECG5000

The ECG5000 dataset (Chen & Keogh, 2000; Goldberger et al., 2000) contains one leadECG data collected from a single patient during a $20\,\mathrm{h}$ monitoring period. While the original dataset (Goldberger et al., 2000) is a single sequence sampled at $100\,\mathrm{samples/s}$, Chen & Keogh (2000) provides $5\,000$ equal-length heartbeats, which were obtained by applying heartbeat segmentation and inter-

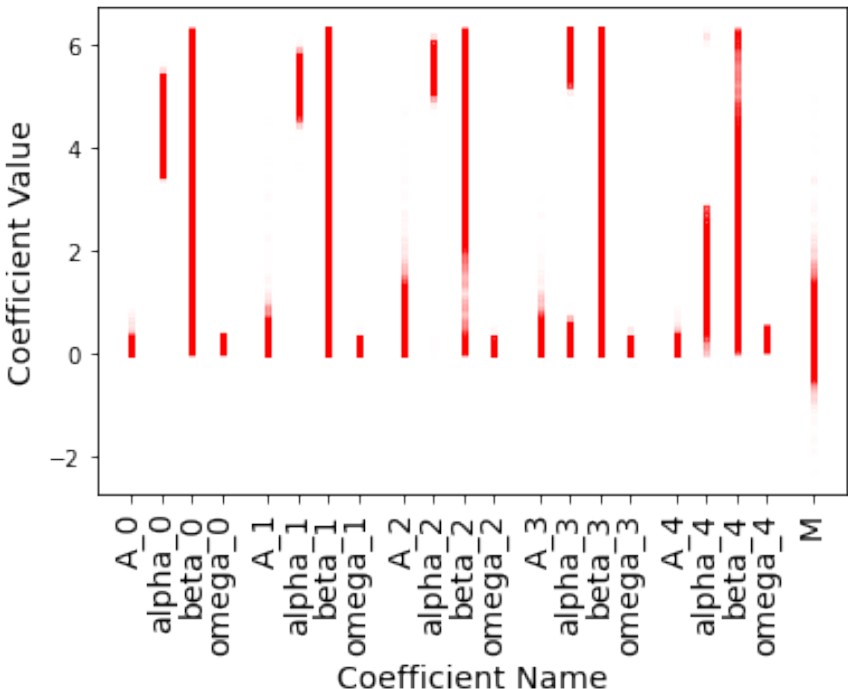

Figure 7: Coefficient distribution for the training set of PTB-XL

polation to 140 final sample lengths. Each heartbeat belongs to one of 5 classes (Normal, Supraventricular, Ventricular, Fusion, and Unclassifiable). The test size is 500 samples, while the remaining 4 500 samples compose the training set size. Although this dataset has been used in numerous previous works, it has numerous constraints that make anomaly detection and classification tasks more straightforward compared to the previously described datasets: $(i)$ the limited number of classes, $(ii)$ all data is from a single patient, and $(iii)$ the constant heartbeat length.

## C.2    MODELS

### C.2.1    ECG-NET

ECG-NET (Roy et al., 2023) is an LSTM-based autoencoder built for anomaly detection of ECG data. The encoder and decoder have a symmetric structure made of 3 layers whose size is progressively decreasing/increasing. The number of units per layer is 128, 64, 32 for the encoder and 32, 64, 128 for the decoder. After each LSTM layer, a dropout (during training) and rectified linear unit (ReLU) function are applied. The dumbbell shape of the model (see Figure 3a) provides a way to compress the original time series into a 32-valued compressed latent space, which is then expanded to reconstruct the signal.

### C.2.2    ENCDECAD

EncDecAD (Malhotra et al., 2016) proposed using a single-layer LSTM-based encoder-decoder to perform anomaly detection on different kinds of time series, drawn from power demand, space shuttle, ECG and engine datasets. During training, knowledge is transferred between the encoder and decoder through the hidden state of the last LSTM cell of the encoder layer, which is used as the initial state for the decoder. In the training phase, the input for the decoder is the original time series, whereas during testing/validation the predicted value from the previous cell is employed. We manually set the size of both encoder and decoder units to 200, which is ~4 times the value (55) used in the original paper for ECG data. We motivate this hyperparameter choice based on the size and complexity of the tested datasets, which are more challenging compared to the dataset utilized in the original paper.

### C.2.3 BERTECG

Alamr & Artoli (2023) proposed employing attention mechanisms (Vaswani et al., 2017) to detect abnormal ECG in two state-of-the-art datasets. We will refer to this model as BertECG, due to its similarity to the widely known BERT (Devlin et al., 2018) transformer for natural language processing. The model includes as the first layer one convolutional embedding layer which translates the input time series to a multi-dimensional embedded vector that can be fed to attention layers. The position information of each timestep in the sequence is added through positional encoding. However, the original encoding is unfit for variable-length heartbeats such as PTB-XL and Shaoxing, since the peaks might be in different positions depending on a patient's heart rate. Hence, we modify the positional encoding structure by mapping position onto the $[0, 2\pi)$ interval and compute the embedded vector consequently.

The core model part includes several attention layers with dropout (rate 0.1) and fully connected layers with ReLU activation, followed by a final linear reconstructing layer. We selected the best-performing hyperparameters from their paper and evaluated the model with 2 layers, 32 attention heads per layer, 128 as model depth, and 128 as the number of units for fully connected layers. The size of the last layer depends on the original signal length.

### C.2.4 CVAE

Motivated by the success of the variational autoencoder (Kingma & Welling, 2022) for anomaly detection and image generation, in Jang et al. (2021) introduce CVAE, a convolutional variational autoencoder for anomaly detection. The model is composed of consecutive blocks, each of them a sequence of one-dimensional convolutional layers, ReLU activation, batch normalization layers, and dropout. Given the numerous parameters and hyperparameters, we refer the reader to their paper and our TensorFlow implementation (Anonymous, 2023) for further information.

To enhance the anomaly detection capabilities, we follow the suggestions from Burgess et al. (2018) and perform a warm-up training as standard non-variational AE (*i.e.,* zeroing out the KL-divergence loss computed for the latent space). Then, we run the actual training with the same weight for KL and reconstruction loss.

### C.2.5 DENSEAE

Additionally, we evaluate a simple baseline, which is composed of 6 fully connected layers with a variable number of units per layer in a dumbbell structure, with ReLU activation followed by a dropout layer. We define the first three layers as an encoder and the last three layers as a decoder.

### C.2.6 FMM-HEAD

We replaced the decoder part of each model with the FMM-Head defined in Section 3. The first fully connected layer has 256 neurons, while the second layer encodes the resulting output to the suitable number of parameters of the input ECG data. Angular coefficients such as $\alpha$ and $\beta$ are encoded into their sine and cosine, while linear variables are encoded by a single output.

### C.3 EVALUATED METRICS

After training, we select the model with the best validation accuracy among all the epochs. As performance metrics, we compute the receiver operating characteristic (ROC) curve and the corresponding area under the curve on an unknown test dataset. We also measure the correlation (either linear or circular for different kinds of extracted coefficients) between the parameters extracted by our FMM-Head enhanced models and those obtained by running the optimization problem(Rueda et al., 2022) with the original R scripts. Firstly, we compare the inference time of our models to the correspondent baselines and the optimization problem. We run exactly the R code provided by Rueda et al. (2022), which extracts the FMM parameters for each single heartbeat sequentially. In case of failure, we discard the sample and do not consider this optimization time in our analysis. To provide a fair comparison, we sequentially compute the inference with our models for one batch of 16 heartbeats at a time. Noteworthy is that our FMM-enhanced models produce explainable coefficients, while the baselines only produce a reconstructed input.

Finally, we compute the model size on disk for the different tested models and compute the ratio between these baselines and our NNs.

## C.4 EXPERIMENTAL SETUP

We ran our code on different kinds of Graphics Processing Units (GPUs), namely NVIDIA L40, A100, and H100. The inference time has been measured for all models on an L40 GPU with CUDA 12.2. The L40 is the least expensive of the three GPUs. The FMM optimization problems were solved on a server running the Ubuntu 20.04 operating system with 2 AMD EPYC 74F3 24-Core Processors with $3\,193.926\,\text{MHz}$ clock frequency, with hyper-treading disabled and one core per optimization problem. Table 3 lists the amount of wall-clock time needed to train the different models for each dataset on an NVIDIA L40 GPU.

Table 3: Training times, including warm-up regression for different models and datasets on an NVIDIA L40 GPU. Since the number of epochs depends on the early-stopping callback, the total training time may be high even for small models.

| Dataset / Model | Training + Warmup time [min] | | |
| --- | --- | --- | --- |
| | Shaoxing | PTB-XL | ECG5000 |
| fmm_bert_ecg | 218.0 | 94.3 | 1.7 |
| bert_ecg | 147.6 | 33.6 | 1.0 |
| fmm_encdec_ad | 59.9 | 29.8 | 1.9 |
| encdec_ad | 28.5 | 12.8 | 2.2 |
| fmm_ecgnet | 156.3 | 70.2 | 3.3 |
| ecgnet | 81.3 | 29.3 | 4.3 |
| fmm_dense_ae | 48.3 | 12.0 | 0.8 |
| dense_ae | 15.3 | 6.4 | 1.2 |
| fmm_cae | 24.9 | 34.3 | 2.6 |
| cvae | 103.2 | 101.6 | 8.4 |

## D ADDITIONAL RESULTS

### D.1 BEST LEARNING RATES

Tables 4 to 6 show the AUROC for multiple AEs including baselines and FMM-Head-enhanced NNs for the Shaoxing, PTB-XL and ECG5000 dataset respectively. Our experiments have been run for 5 times for each combination of dataset, model and learning rate. We report the average and standard deviation of the AUROC value.

Table 4: AUROC values for different models and learning rates on Shaoxing dataset

| Lr / Model | 0.00001 | 0.00005 | 0.00010 | 0.00050 | 0.00100 |
| --- | --- | --- | --- | --- | --- |
| ecgnet | $0.575 \pm 0.007$ | $0.574 \pm 0.006$ | $0.568 \pm 0.012$ | $0.557 \pm 0.011$ | $0.566 \pm 0.013$ |
| fmm_ecgnet | $0.660 \pm 0.008$ | $0.659 \pm 0.008$ | $0.651 \pm 0.006$ | $0.641 \pm 0.006$ | $0.642 \pm 0.004$ |
| encdec_ad | $0.461 \pm 0.003$ | $0.461 \pm 0.003$ | $0.460 \pm 0.003$ | $0.460 \pm 0.003$ | $0.462 \pm 0.006$ |
| fmm_encdec_ad | $0.618 \pm 0.004$ | $0.609 \pm 0.008$ | $0.613 \pm 0.006$ | $0.603 \pm 0.011$ | $0.607 \pm 0.014$ |
| bert_ecg | $0.535 \pm 0.008$ | $0.512 \pm 0.021$ | $0.510 \pm 0.041$ | $0.546 \pm 0.016$ | $0.538 \pm 0.050$ |
| fmm_bert_ecg | $0.644 \pm 0.004$ | $0.645 \pm 0.005$ | $0.651 \pm 0.004$ | $0.612 \pm 0.040$ | $0.560 \pm 0.002$ |
| dense_ae | $0.636 \pm 0.004$ | $0.651 \pm 0.005$ | $0.653 \pm 0.004$ | $0.650 \pm 0.004$ | $0.637 \pm 0.003$ |
| fmm_dense_ae | $0.669 \pm 0.006$ | $0.678 \pm 0.007$ | $0.684 \pm 0.008$ | $0.699 \pm 0.006$ | $0.692 \pm 0.003$ |
| cvae | $0.698 \pm 0.004$ | $0.718 \pm 0.007$ | $0.730 \pm 0.010$ | $0.749 \pm 0.022$ | $0.737 \pm 0.013$ |
| fmm_cae | $0.713 \pm 0.007$ | $0.729 \pm 0.006$ | $0.727 \pm 0.006$ | $0.679 \pm 0.057$ | $0.563 \pm 0.004$ |

Table 5: AUROC values for different models and learning rates on PTB-XL dataset

| Model \ Lr | 0.00001 | 0.00005 | 0.00010 | 0.00050 | 0.00100 |
|---|---|---|---|---|---|
| ecgnet | 0.662 ± 0.011 | 0.599 ± 0.055 | 0.578 ± 0.049 | 0.541 ± 0.054 | 0.575 ± 0.034 |
| fmm_ecgnet | 0.701 ± 0.007 | 0.732 ± 0.005 | 0.731 ± 0.003 | 0.718 ± 0.004 | 0.712 ± 0.005 |
| encdec_ad | 0.381 ± 0.000 | 0.384 ± 0.006 | 0.381 ± 0.000 | 0.382 ± 0.002 | 0.382 ± 0.002 |
| fmm_encdec_ad | 0.695 ± 0.008 | 0.689 ± 0.003 | 0.677 ± 0.003 | 0.653 ± 0.002 | 0.643 ± 0.007 |
| bert_ecg | 0.520 ± 0.032 | 0.489 ± 0.061 | 0.471 ± 0.031 | 0.499 ± 0.052 | 0.537 ± 0.030 |
| fmm_bert_ecg | 0.697 ± 0.005 | 0.694 ± 0.003 | 0.673 ± 0.021 | 0.673 ± 0.005 | 0.676 ± 0.013 |
| dense_ae | 0.673 ± 0.006 | 0.683 ± 0.004 | 0.685 ± 0.003 | 0.691 ± 0.003 | 0.692 ± 0.003 |
| fmm_dense_ae | 0.660 ± 0.001 | 0.671 ± 0.003 | 0.683 ± 0.003 | 0.698 ± 0.002 | 0.699 ± 0.009 |
| cvae | 0.694 ± 0.003 | 0.683 ± 0.005 | 0.684 ± 0.006 | 0.667 ± 0.003 | 0.663 ± 0.014 |
| fmm_cae | 0.710 ± 0.007 | 0.722 ± 0.003 | 0.718 ± 0.004 | 0.727 ± 0.007 | 0.649 ± 0.076 |

Table 6: AUROC values for different models and learning rates on ECG5000 dataset

| Model \ Lr | 0.00001 | 0.00005 | 0.00010 | 0.00050 | 0.00100 |
|---|---|---|---|---|---|
| ecgnet | 0.983 ± 0.002 | 0.993 ± 0.001 | 0.994 ± 0.001 | 0.992 ± 0.001 | 0.993 ± 0.001 |
| fmm_ecgnet | 0.977 ± 0.012 | 0.983 ± 0.006 | 0.988 ± 0.005 | 0.987 ± 0.003 | 0.988 ± 0.001 |
| encdec_ad | 0.927 ± 0.000 | 0.964 ± 0.032 | 0.982 ± 0.006 | 0.982 ± 0.001 | 0.982 ± 0.005 |
| fmm_encdec_ad | 0.960 ± 0.008 | 0.976 ± 0.006 | 0.965 ± 0.032 | 0.988 ± 0.003 | 0.988 ± 0.003 |
| bert_ecg | 0.963 ± 0.006 | 0.949 ± 0.020 | 0.962 ± 0.011 | 0.962 ± 0.008 | 0.971 ± 0.008 |
| fmm_bert_ecg | 0.982 ± 0.007 | 0.981 ± 0.009 | 0.986 ± 0.006 | 0.989 ± 0.001 | 0.988 ± 0.002 |
| dense_ae | 0.986 ± 0.002 | 0.993 ± 0.001 | 0.993 ± 0.001 | 0.992 ± 0.000 | 0.992 ± 0.001 |
| fmm_dense_ae | 0.979 ± 0.004 | 0.984 ± 0.006 | 0.987 ± 0.005 | 0.990 ± 0.002 | 0.990 ± 0.002 |
| cvae | 0.993 ± 0.000 | 0.993 ± 0.001 | 0.993 ± 0.001 | 0.993 ± 0.001 | 0.992 ± 0.001 |
| fmm_cae | 0.969 ± 0.006 | 0.985 ± 0.005 | 0.986 ± 0.005 | 0.990 ± 0.000 | 0.990 ± 0.002 |

## D.2 WARM-UP REGRESSION

Figure 10 shows a normally labeled ECG reconstruction for the ECG5000 dataset where warm-up was *not* applied before training for anomaly detection. We can see that the two extracted waves on the right both contribute to the ending section of the ECG wave (*i.e.,* the T peak). Moreover, only two extracted waves can be seen in the QRS complex, instead of three. Hence, there is no clear distinction between waves and thus the impact of the FMM-Head on the anomaly detection capabilities is attenuated.

As a positive example, Figure 8 depicts the FMM waves after warm-up regression and training on the PTB-XL dataset with the FMM-Head integrated in DenseAE. We compare the results for the same sample when warm-up was *not* applied (Figure 9). In (b) the obtained waves are random, since without regression the NN does not learn the order of the peaks. Also, only two waves compose the QRS complex, because the last coefficients produce a minor wave after the T peak (on the right). The relevance of warm-up is emphasised for ECG5000 (Figure 10), where the five waves are not clearly distinguishable, expecially in the QRS complex where the peaks are close.

## D.3 ADDITIONAL RESULTS FOR PTB-XL

Figures 4b, 12 and 13 show respectively the inference times, model size, and epoch times for our models and the baselines on the PTB-XL dataset. The results are similar to those for Shaoxing. However, the inference times for LSTM-based layers are higher in Shaoxing since the input time series are longer. This is due to the fact that the sampling frequency is $500\,\mathrm{samples/s}$ for Shaoxing and $100\,\mathrm{samples/s}$ for PTB-XL.

As shown in fig. 14 and similarly to Shaoxing, the correlation between the original FMM coefficients and the one predicted by FMM-Head is higher for the P and T wave compared to the QRS complex. The only exception is the $\omega$ parameter of the Q wave, whose correlation is negligible for the PTB-

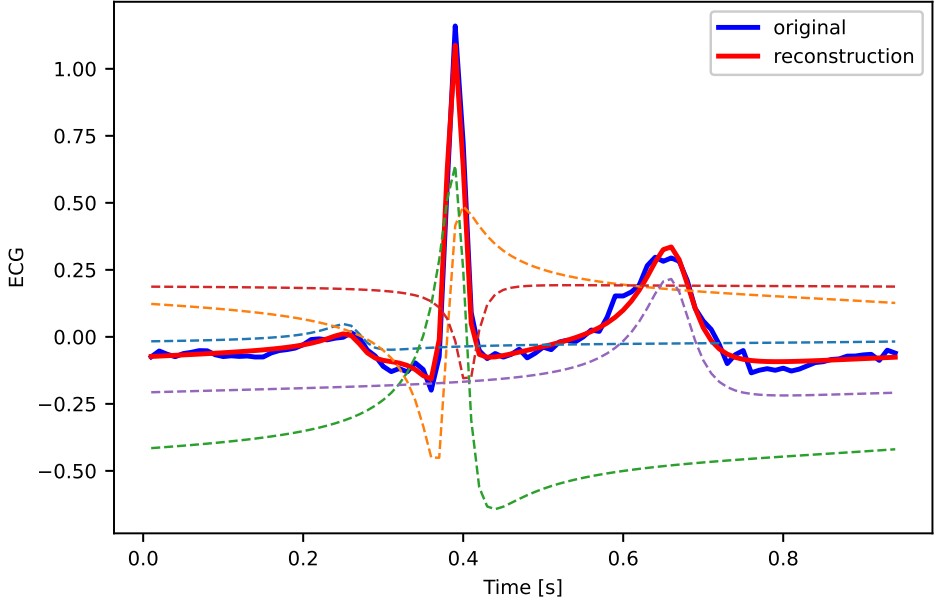

Figure 8: Warm-up on PTB-XL

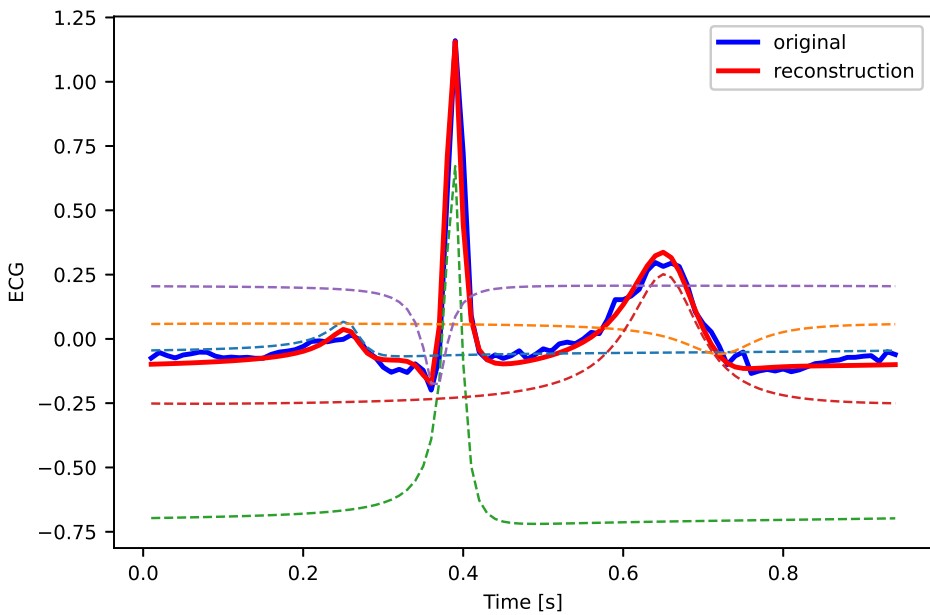

Figure 9: No warm-up on PTB-XL

XL dataset. One cause of this behavior is the activation function for the $\omega$ parameter in FMM-Head that constrains it between 0 and $\omega_{\max}$, thus affecting the correlation to the original parameter.

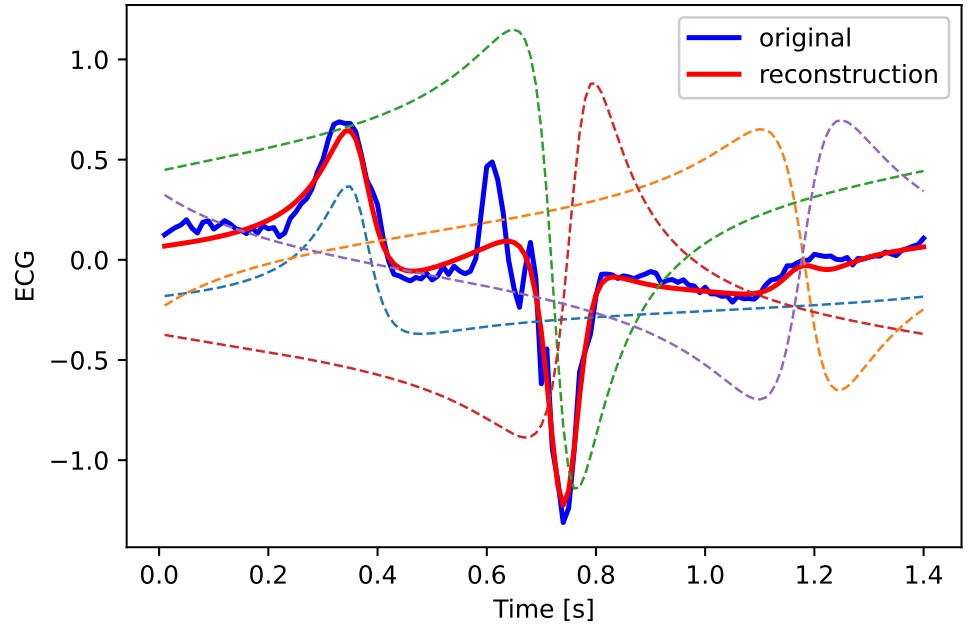

Figure 10: No warm-up on ECG5000

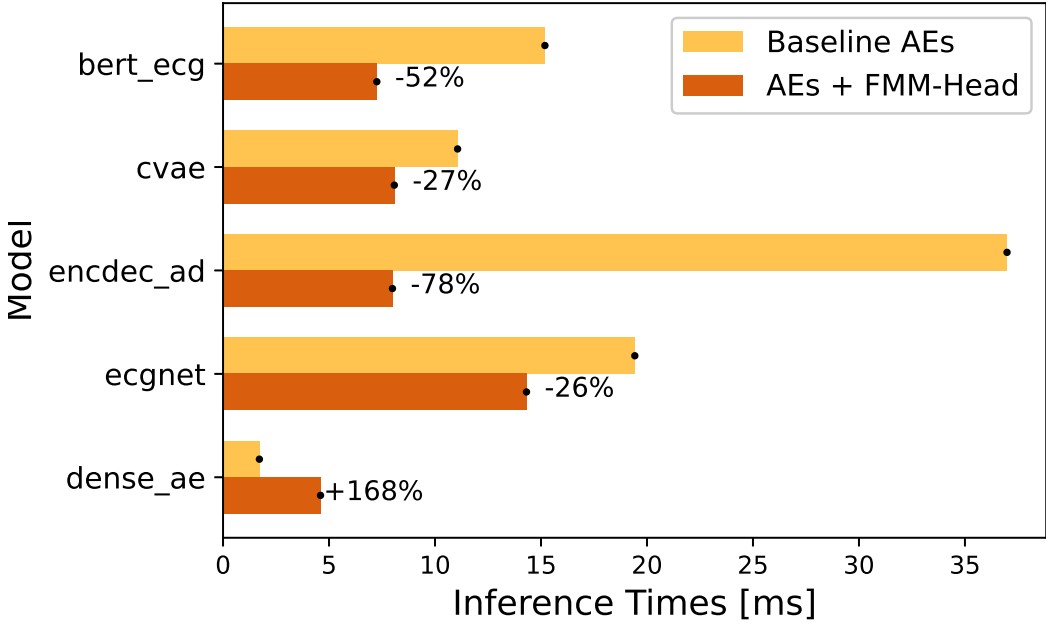

Figure 11: Inference time for PTB-XL

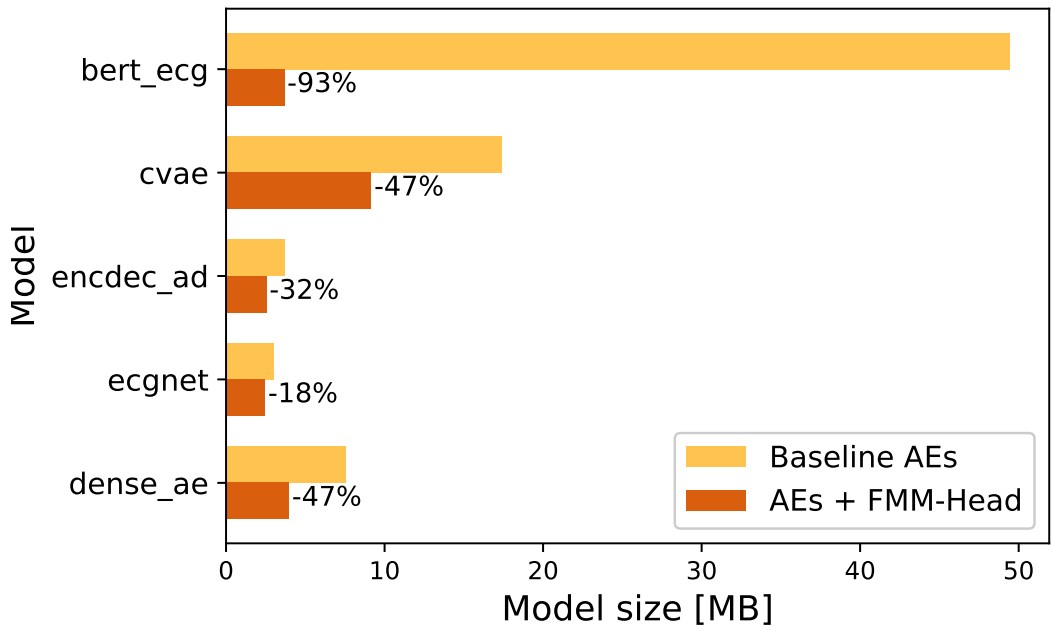

Figure 12: Model size for PTB-XL

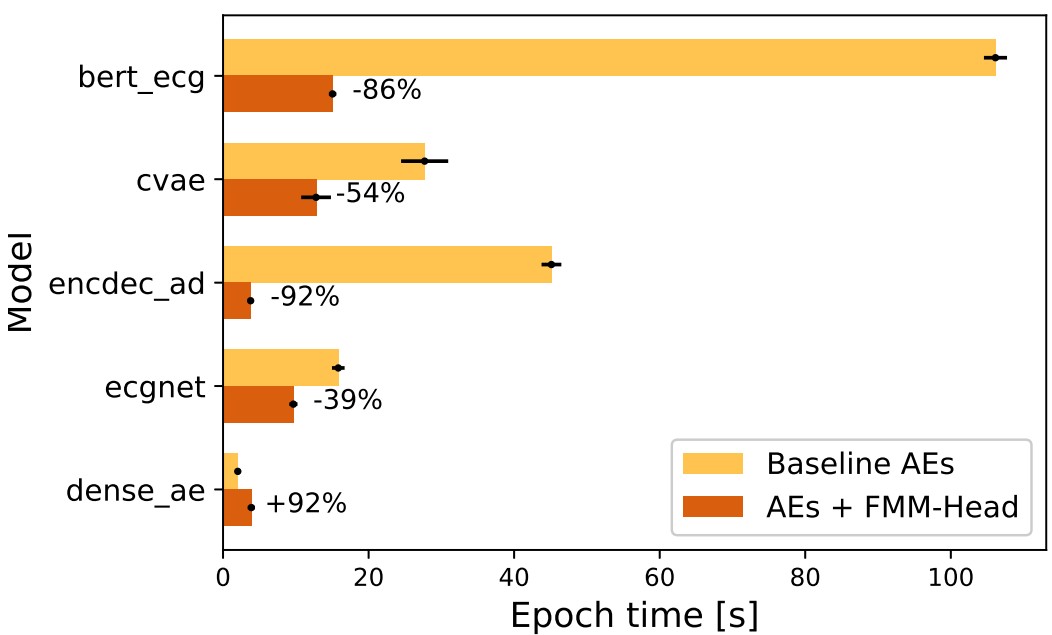

Figure 13: Train epoch times for PTB-XL

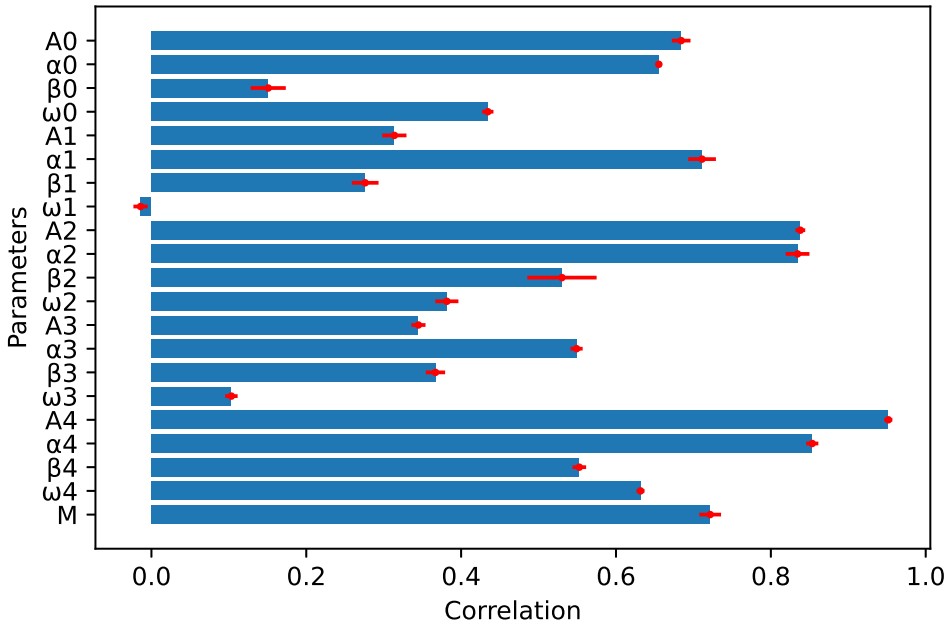

Figure 14: FMM coefficients correlation for PTB-XL

