# OpenReview forum: "FMM-Head: Enhancing Autoencoder-based ECG anomaly detection with prior knowledge"
_ICLR.cc/2024/Conference — ICLR 2024 Conference Withdrawn Submission_

### Official Review · Reviewer_2MCh · 2023-10-16

**Soundness:** 3 good
**Presentation:** 2 fair
**Contribution:** 3 good
**Rating:** 5
**Confidence:** 4

**Summary:**

Inspired by FMM, the authors devised an FMM-HEAD for ECG anomaly detection. To accomplish this, various strategies, such as warm-up regression and some formulas for variable range requirements, are incorporated. Through experimentation on three datasets, it is substantiated that the model not only outperforms its counterparts but also operates at a lower computational cost.

**Strengths:**

- The paper's primary strength lies in its adept transformation of recent breakthroughs within the cutting-edge field of medical research into deep learning field/approach. This signifies a highly commendable approach to advancing AI research for healthcare.

- The performance is good, and the computational complexity is reduced compared to prior research efforts.

**Weaknesses:**

- The presentation requires improvement, as many crucial details are missing. In a specialized and interdisciplinary field like AI for healthcare, readers may not possess knowledge about concepts or information that has not been explicitly introduced.

- The illustrations are quite haphazard and lack standardization. For instance, in the case of Figure 3 (see questions below). See Figure 2, the author even used variable names from the code, which makes reading this paper highly inconvenient.

- I believe that the FFM-head may not always yield benefits across all types of heart diseases. Could the authors provide a theoretical analysis and summary about which kinds of heart diseases are best suited for this approach and which ones are not? Additionally, it would be valuable to investigate whether FMM exhibits particular advantages in scenarios where training data is limited, and whether the performance gains become insignificant when an ample training set is available.

- There must be some DL models for anomaly detection that is not AE-based. It is important to compare with them and observe if we really need to use the FMM-head.

**Questions:**

- You should remove those useless reference on some terms like "machine learning (ML)", "ECG", "AE"...

- Parts of the citation formats are wrong. e.g., EncDecAD in the experiment section.

- Figure 3 is unclear, especially the right part. There are so many arrows, but it is hard to know their meaning.

- How to implement "Circular-to-Linear" is not given.

- It is not clear how to do anomaly detection by AE. Though authors said "AEs are state-of-the-art models for anomaly detection (Chalapathy & Chawla, 2019). AEs are trained on normal data so that abnormal samples during test phases can straightforwardly be recognized." But it is still not clear for readers how this works. You may give something concrete (e.g., by equation), especially similar operations are exerted on your model.

- Given that FMM is a recently introduced methodology, I am uncertain about its widespread acceptance within the medical community and whether subsequent research has substantiated its efficacy. This uncertainty also extends to the potential value of the FMM-HEAD.

---

### Official Review · Reviewer_kmWJ · 2023-10-29

**Soundness:** 2 fair
**Presentation:** 2 fair
**Contribution:** 2 fair
**Rating:** 3
**Confidence:** 5

**Summary:**

This paper proposes a novel neural network model called FMM-Head for solving the ECG abnormality detection problem. The FMM-Head model employs a novel decomposition architecture that replaces the decoding part of the self-encoder with a reconstructed head based on the a priori knowledge of the shape of the ECG. Compared with the traditional self-encoder model, the FMM-Head model performs well in ECG abnormality detection with an improvement of 0.311 in the area under the ROC curve (AUROC).

**Strengths:**

1. The motivation is clear.
2. The model has interpretable extracted features and small model size for real-time ECG parameter extraction and abnormality detection.
3. The paper provides a detailed analysis of the performance and interpretability aspects of the model.

**Weaknesses:**

1. The core idea is trivial. More baselines are needed for example GAN-based methods and diffusion-based methods for ECG data.
2. It is strange to use “anomaly detection” for ECG related task.
3. More experimental validation can be performed to demonstrate the robustness and scalability of the model.
4. More in-depth analysis should be conducted in this paper to explore the limitations and optimization space of the model.

**Questions:**

Please see above

---

### Official Review · Reviewer_YaBZ · 2023-10-30

**Soundness:** 3 good
**Presentation:** 4 excellent
**Contribution:** 2 fair
**Rating:** 3
**Confidence:** 5

**Summary:**

The authors propose to replace the decoder part of the auto-encoder by FMM-Head, composition of 5 waves to mimic normal human ECG, for anomaly detection. The capability of anomaly detection was well tested.

**Strengths:**

To replace the decoder part of the auto-encoder by FMM-Head is quite an interesting idea for domains like ECG where data are indirect observation of some physiological activity. The approach sounds, and well tested within the scope of abnormal detection.

**Weaknesses:**

Abnormal detection is just an initial analysis of ECG data and there are many ML models that can classify normal and 5 different arrhythmias etc. Thus the scope of the paper is narrow for both ML for ECG as well as the general audience of ICLM. It would be better to discuss in a more specific venue.

**Questions:**

The datasets used have multiple classes in abnormal ECG. Have you tested classification tasks for those diagnostic labels?

---

### Official Review · Reviewer_johH · 2023-10-31

**Soundness:** 2 fair
**Presentation:** 2 fair
**Contribution:** 1 poor
**Rating:** 1
**Confidence:** 5

**Summary:**

The paper proposes a method for ECG anomaly detection. The method replaces the decoding part an AE with a reconstruction head called FMM-Head that uses prior knowledge of the ECG shape. Experiments are done on 2 datasets: PTB-XL and ECG5000. 5 baseline models are used for implementing the new decoder namely ECG-NET, EncDecAD, BertECG, CVAE, and a fully connected AE. The results show improvements over standard decoders.

**Strengths:**

The paper is situated in an important problem area. The method is simple and easy to understand.

**Weaknesses:**

Following are the weaknesses of the paper:

- The focus of the problem statement is too narrow. While the general area of anomaly detection is broad and important, the method only explores AE setups. It is not clear how relevant and important this is since AEs are not the only way of ECG anomaly detection. There are also no indications as to how generalizable this solution is.

- The paper claims that existing models "do not consider the specific patterns of ECG leads and are unexplainable black boxes". I would like to challenge both notions. First, it is completely possible that the ECG patterns and PQRST waveforms are implicitly learned by existing methods. There is no evidence that they are "not considered". It would be hard to imagine how prior works produce valid results without learning the waveforms. Second, the "black box" argument is not well-founded. There are many studies that have analyzed and interpreted the performance of other methods.

- The writing can be improved by removing very simple notions such as Pearson's correlation (page 5) and other well-known concepts.

- There seems to be a gap in the logic of the paper. The paper states "The parameters obtained after the activation function are used to reconstruct an ECG time-series Xˆ for anomaly detection. We leverage this weak a priori knowledge drawn from the FMM formulation to better reconstruct the input ECG signal." How is this considered 'a priori knowledge'? and why is it 'weak'? How is this different from using a standard decoder in terms of reconstruction based extracted features of one form or another?

- The experiment setup requires substantial improvements as only 5 AE-based methods are used. Are these the only methods that can be used for ECG anomaly detection? what about non-AE methods?

**Questions:**

Please see my comments under weaknesses.

---

### Author Response · Authors · 2023-11-22
**Comment to reviewers**

The authors would like to thank the reviewers for their insightful comments. Unfortunately, due to the limited amount of available time we were not able to fully implement and test additional models such as GANs and diffusion-based techniques. We are currently working on addressing the comments, but we will not be able to do that during the review process.
Kind Regards,
The authors